# The self-management work of food hypersensitivity

**Monika Dybdahl Jakobsen**[1,2]\*, **Aud Obstfelder**[3], **Tonje Braaten**[2], **Birgit Abelsen**[4]

**1** Center for Care Research, UiT The Arctic University of Norway, Tromsø, Norway, **2** Department of Community Medicine, UiT The Arctic University of Norway, Tromsø, Norway, **3** Center for Care Research, The Norwegian University of Science and Technology (NTNU), Gjøvik, Norway, **4** Department of Community Medicine, Norwegian Centre of Rural Medicine, UiT The Arctic University of Norway, Tromsø, Norway

\* monika.d.jakobsen@uit.no

## Abstract

### Background

Food hypersensitivity (FH) has received considerable attention in the scientific community in recent years. However, little attention has been given to the efforts people make to manage their FH. We aimed to explore these efforts by using Normalization Process Theory, which is a conceptual framework formerly used to describe the self-management 'work' of long-term conditions.

### Methods

We carried out qualitative individual interviews with 16 women with FH. Transcripts from recorded interviews were analyzed using template analysis.

### Results

16 women participated; some had diagnoses from conventional medicine (celiac disease, lactose intolerance, food allergies, irritable bowel syndrome) and some did not. Participants described carrying out several tasks, some of which were time-consuming, to manage their FH. Women who had clarified once and for all what food(s) caused symptoms, described that they could concentrate on carrying out a restricted diet, which could become routine. Conversely, participants who had not achieved such clarification described carrying out tasks to identify what food(s) caused symptoms, and to implement and evaluate a tentative diet. Participants' descriptions also revealed a heightened vigilance when they ate food that others had prepared, and some made efforts to conceal their FH.

### Conclusions

Self-management of FH may, like the self-management of other long-term conditions, imply a large workload and burden of treatment. Efforts made to conceal FH may be considered part of this workload, while help in clarifying which food(s) cause symptoms has the potential to reduce the workload.

**Data Availability Statement:** Data cannot be shared publicly because the qualitative interviews contain potentially identifying and sensitive patient information, and because the ethics approval for this study and informed consent from participants did not request for permission to make any parts of

transcripts, except anonymized quotations used in study outputs, publicly available. Norwegian centre for research data, which has not approved public access to the data, can be contacted at the following email address: nsd@nsd.no. The data underlying the results presented in the study can be found on Services for sensitive data, https://www.uio.no/english/services/it/research/sensitive-data/, and are available on reasonable request.

**Funding:** MJ was supported by the Department of Community Medicine, UiT The Arctic University of Norway (https://en.uit.no/enhet/ism), grant agreement 4025. The funders had no role in study design, data collection and analysis, decision to publish, or preparation of the manuscript.

**Competing interests:** The authors have declared that no competing interests exist.

## Introduction

Food hypersensitivity (FH) is a collective term for adverse reactions to foods, also referred to as food allergies and food intolerances [1]. FH has received considerable attention in the media and in the scientific community in recent years [2]. Existing studies on FH have focused on several topics. Some have investigated epidemiological aspects of FH, such as the prevalence of FH [3,4], while others have explored dietary and diagnostic aspects of FH [5,6]. Some qualitative studies have explored the experiences of people who live with FH, including the experiences of eating out [7,8], and some have explored the quality of life of people with different FH conditions [9–17]. However, although FH in adults is usually a long-term condition with no treatment and thus must be managed in everyday life by avoiding or reducing the ingestion of foods that cause adverse reactions [18], little attention has been given the efforts people make to manage their FH.

Previous studies have explored the efforts people make to manage other long-term conditions, such as diabetes, respiratory diseases, cardiovascular disease, and musculoskeletal disorders [19–22]. In these studies, the effort used and the tasks carried out to self-manage the conditions, i.e., to avoid exacerbation events and recurrence, are referred to as 'work' [19–21,23]. In addition, the self-management workload and its effect on functioning and well-being are referred to as the 'burden of treatment' [23–25]. In recent years, an increasing amount of studies have investigated self-management work and burden of treatment [19–21,24,26], and the literature suggests that self-management work can entail a large workload, can occupy a large part of the day, and may limit the time persons can spend on other activities, such as occupational work [19,21,25]. In addition, the self-management workload may overburden patients, particularly those with multiple conditions [23,25]. It has been argued that being overburdened can lead to poor adherence to recommended treatments, and that health care providers should therefore be attentive to peoples' total burden of treatment and strive to keep the self-management workload feasible for persons with long-term conditions [23,25,27,28].

Some studies have employed the Normalization Process Theory (NPT) to explore the self-management work of long-term conditions [19,20]. The conceptual framework NPT encompasses four core constructs, which have been used to categorize the self-management work of long-term conditions [19,20]. The first of these constructs, 'coherence', includes the work carried out in order to understand the condition, its consequences, and its treatment. The second construct, 'collective action', refers to the implementation and execution of concrete self-management tasks, such as taking treatments or enacting lifestyle changes. The third construct, 'cognitive participation', includes the work related to engaging others in the management of the long-term condition, engaging others to provide support, and arranging help from others. The fourth construct, 'reflexive monitoring', refers to the evaluation and alteration of the treatment regimen [19,20]. Gallacher et al. [19] developed an operationalization of NPT which can be used as a tool to explore and categorize the self-management work of long-term conditions. In the present study, we aimed to explore the efforts people make to manage their FH, using Gallacher et al.'s [19] operationalization of the NPT as a tool to explore and describe these efforts.

## Materials and methods

### Design and sample

In order to explore the efforts people make to manage their FH, semi-structured, qualitative individual interviews with 16 women with FH aged 39–67 years were carried out. The

inclusion of women in this age span was a result of the fact that the present study is part of a project that only includes women from a certain age span. People with different FH diagnoses based on conventional medicine tests, such as celiac disease, lactose intolerance, or food allergies [18,29,30], those with symptom-based diagnoses from conventional medicine, such as irritable bowel syndrome [31], and those with self-diagnosed FH may all make efforts to manage their FH. Therefore, and in order to ensure variation, we included participants from each of these groups. More specifically, we included eight women with different FH diagnoses based on conventional medicine tests, four women with irritable bowel syndrome, and four women without a FH diagnosis from conventional medicine.

The choice to recruit 16 interviewees was influenced by Guest et al. [32], who argues that 12 interviewees are usually sufficient to achieve data saturation in qualitative samples, but that a somewhat larger sample can be considered if the sample is heterogeneous. Since the sample in the present study was heterogeneous with regard to FH diagnoses, we chose to recruit slightly more than 12 interviewees.

Most participants were recruited through contacts in The Norwegian Asthma and Allergy Association and Norwegian Celiac Association, or through acquaintances of the first author, using the following procedure: The first author asked the contacts whether they knew women with FH in the relevant age group who might want to participate in the study. The contacts then got in touch with possible participants, gave them information about the study, and asked whether they wanted to participate in the study. If a woman wanted to participate, the first author received her name and phone number from the contacts, and called the woman to arrange an interview. In addition, one woman self-recruited when she heard about the study topic, and one woman was recruited though an invitation on The Norwegian Asthma and Allergy Association's Facebook site.

The interviews were conducted face-to-face, and only the interviewer and the interviewee were present. The interviews were carried out in one of the following undisturbed locations, which was chosen by the interviewee: the participant's place of work, at the participant's home, or on the premises of UiT The Arctic University of Norway. The interviews took place in five different towns in Norway, lasted 53–98 minutes (mean 67 minutes), and were carried out from August until November 2016. All interviews were carried out in the Norwegian language. The interviewer and 15 of the participants were native Norwegian speakers, while one participant spoke fluent Norwegian.

Before the project started, a remit assessment was sent to the Regional Committee for Medical and Health Research Ethics (REC) North. REC North responded that this project did not require approval from them (2014/1565). After this, in line with procedures, the Norwegian Centre for Research Data (Project number 40138) was notified of the project. All women gave written informed consent before they participated in the study.

## Data collection and analysis

The first author conducted the interviews and carried out the analysis in close collaboration with the other authors. Gallacher et al.'s [19] operationalization of the NPT was used as a tool to explore the self-management work of FH, thus the initial template and the associated interview themes and interview guide were inspired by Gallacher et al.'s [19] operationalization of the NPT (See Table 1 for initial template). Since this operationalization of the NPT offers predefined themes, we chose to carry out a template analysis, which is a style of thematic analysis that allows the use of themes that are determined in advance of data collection [33,34]. In template analysis, the initial themes are usually applied to parts of the data, and where the initial themes do not fit the data, the template is modified [33,34]. The modified template is then applied to the full data

**Table 1. The initial and final template.**

| Initial template | Final template |
|---|---|
| *Coherence (sense-making work)* | *Coherence (sense-making work)* |
| Understanding the food hypersensitivity. | (similar to the initial template) |
| How they gained information about the food hypersensitivity and how it could be managed. | |
| Own understanding compared to information achieved. | |
| A clear picture of the food hypersensitivity? | |
| *Collective action (enacting work)* | *Collective action (enacting work)* |
| Setting a strategy to cope with symptoms. | (similar to the initial template) |
| Executing the strategy. | |
| Making sure they have the right resources. | |
| *Cognitive participation (relationship work)* | *Cognitive participation (relationship work)* |
| Engaging others to provide support. | What they do when sharing meals with others/ eating food others have served. |
| Arranging help. | |
| Legitimation. | Legitimation. |
| *Reflexive monitoring (appraisal work)* | *Reflexive monitoring (appraisal work)* |
| Altering routines? | (similar to the initial template) |
| Discussing or altering management plans with others. | |
| Assessing individually whether to alter plans. | |
| Keeping up to date. | |
| | *Concealment work* |

[33,34]. In the present study, we tested the initial themes and the associated interview guide in a pilot interview before we used them on the first study participants. The pilot interview was conducted with a woman with FH, using the same procedures as those applied in the following participant interviews. However, the pilot interview was not audio recorded. While the pilot interview led to small modifications of the interview guide, findings from the first study interviews led to larger modifications of the interview guide and template, and these modifications were as follows: 1) The first interviews revealed that some participants made efforts to conceal their restricted diet and FH. We interpreted these efforts as part of the work of handling FH; we were attentive to this in the rest of the interviews, and included the topic in the final coding template. 2) While the initial interview guide, in line with Gallacher et al. [19], included questions concerning help from a person's social network and health care services, the participants in the present study gave few examples of such help. Consequently, the interview guide and template were changed in line with this observation. More concretely, themes concerning help from others were replaced by participants' practices when sharing meals with others (see Table 1 for final template).

All interviews were audio recorded and transcribed by a professional transcriber. The analysis was as follows: after each interview, the interviewer (first author) wrote notes, which were discussed with the coauthors, and which led to the aforementioned modifications of the interview guide. After all interviews were completed, the first author read through the transcriptions to become familiar with the data, before she read through each transcription again and noted the descriptions of all tasks carried out and efforts made in order to manage FH. These tasks and efforts were interpreted as the self-management work of FH, were categorized as per the final coding template, and were discussed in the research team to identify characteristics of the self-management work of FH, patterns and differences, as well as relationships between different types of work. After this, the first author searched the transcriptions to find quotes that best illustrated the themes. These quotes were translated from Norwegian to English, with

an emphasis on preserving the contents of the participant's statement, but also with the aim to give a translation that was as direct as possible.

## Results

### Participant characteristics

The 16 women we recruited were aged 39–67 years (mean age 49.7 years). More specifically, two women were 39 years of age, five were in the age group 40–49, seven were in the age group 50–59, and two were in the age group 60–67 years. All women described having had FH for a long time, i.e., for years or decades. While some of these women described experiencing symptoms of their FH very recently, other had managed to avoid these symptoms for months or years. At the time of recruitment, before the interviews, they reported the following FHs: three had celiac disease, one had lactose intolerance, one had both celiac disease and lactose intolerance, and three had food allergies. In addition, four had irritable bowel syndrome, and four participants did not have a FH diagnosis from conventional medicine. However, during the interviews, most participants with celiac disease, lactose intolerance, or food allergies described that they also had irritable bowel syndrome or an additional, self-diagnosed FH. Most of the 16 women described that they were hypersensitive to more than one food, and many described being hypersensitive to foods and components that are included in many products and meals, such as gluten, wheat, and milk.

Among the 16 participants, four women had secondary school as their highest completed education level, while six had a bachelor's degree, and six had a master's degree. Nine women had minor children, and eight of these women lived with a partner. Seven women had adult children, and six of these lived with a partner. Eight of the women had other long-term conditions, such as diabetes, hypothyroidism, rheumatism, fibromyalgia, asthma, or allergies. Participants also described that they had experienced periods of poor health and/or exhaustion as a result of poorly managed FH.

### Coherence: Identifying which foods cause symptoms and what concrete products to eat

All women described some kind of initial phase, during which they experienced symptoms like diarrhea, stomach pain, nausea, low energy, weight loss, breathing problems, or skin symptoms. These symptoms lasted for a long time, and most women described considerable symptoms. In order to find out what caused these symptoms, the women reported carrying out different tasks. Some contacted health care services and took a celiac disease test, a lactose intolerance test, and/ or allergy tests. However, several women said that they received little diagnostic help from health care services. Some participants had searched the internet, read books, contacted alternative medicine clinics, and/or removed and reintroduced foods. One participant without a FH diagnosis from conventional medicine described such a reintroduction process:

*I was supposed to test it out; eat a tablespoon or so of a potato. So I would do that for 3 days and then wait and see. Then I would do the same thing with a new food. It was a kind of rotation diet, and it was a cautious approach to it. I continued this for a long time.*

Some of the women described having done several of the above-mentioned tasks without finding out what caused symptoms. One study participant with irritable bowel syndrome explained why she had done so many different tasks to find out what caused symptoms:

*You search high and low when you are as afflicted as I have been. You have to search, because [the condition] influences the quality of your life very much.*

While women with a diagnosis of celiac disease from conventional medicine described they received information from health care services about what concrete products and meals to eat, participants without a celiac disease diagnosis said they had to find this information themselves. In particular, those who were hypersensitive to many foods and to foods that are present in common products and meals, described that this could take a long time and could be challenging. Some also described finding out what products and meals to eat as so demanding that they tended to eat only a few, familiar, 'safe' dishes. One study participant with lactose intolerance verbalized the process of finding out what to eat in the following way:

*In the beginning, it was a lot of finding out what I could use instead. (. . .) I found out that here in Norway there is milk in everything (. . .) in stuff you could not dream that there would be milk in.*

## Collective action: Carrying out a restricted diet

In general, the women said that they spent more time shopping for groceries and preparing food now than they did before they implemented a restricted diet. They read ingredient labels consistently and made more food from scratch. Some searched the internet for recipes, and/or used weekends or evenings to prepare food for the next days. One study participant without a FH diagnosis from conventional medicine reported spending 1 hour each evening to make herself food for the next day:

*You have to find time [to make food]. (. . .) I make [food for the next day] once the children have gone to bed. (. . .) It takes me approximately 1 hour. (. . .) I have to; I do it because it keeps me healthy, and I do not want to go back to where I was. I want to work, I want to stay healthy.*

## Cognitive participation: Eating with others

As previously mentioned, the participants described that they carried out several tasks, some of which were time-consuming, to manage their FH. These descriptions included many examples of how they carried out these tasks by themselves, and few examples of help from others. The women also described that they would watch while others prepared food, because they had learned this was necessary, and that they read ingredient labels on groceries others had bought. One participant with celiac disease explained how she always washed her kitchen before she started making her own food:

*Yes, because at home it is worse, because they forget. (. . .) I have to wash everything. (. . .) In fact, my risk is highest at home.*

The women presented different strategies related to eating at friends' homes, restaurants, cafes, etc. Some of these strategies included letting the hosts know in advance about their FH, asking their husbands to taste food first, bringing their own food, and picking out foods they could tolerate. Some clarified the contents of food with the chef or waiter, and they described that they often had to double or triple check these answers.

The women's descriptions revealed a heightened vigilance when they ate food that others had prepared. One participant without a FH diagnosis from conventional medicine described eating food others had prepared with the following metaphor:

*Compared to the many who do not have these issues, I feel like I have a computer that runs in the background and processes data continuously. (. . .) at a meal that I do not control 100%, I must somehow scan and think through. If I sometimes choose to eat something that I am unsure of (. . .) there may be consequences.*

### Reflexive monitoring: Evaluation of the self-management regimen

Participants who had clarified once and for all which food(s) caused symptoms, gave few descriptions of how they evaluated their self-management regimens. These women mainly gave descriptions of how they carried out their restricted diet, and for those who had carried out the same food restrictions for months or years, shopping and preparing food had become routine. This group included women who had only one FH condition that had been diagnosed by conventional medical tests, such as participants with celiac disease only. On the other hand, women who had not clarified once and for all which food(s) caused symptoms, described a cycle of repeated tasks to find out what caused symptoms, implement new tentative diets, instruct others in tentative diets, and evaluate these tentative self-management regimens (Fig 1). This group included women with several FH conditions and women with conditions that were not diagnosed by conventional medicine.

### Concealment work

Both participants with different FH diagnoses from conventional medicine and participants without such diagnoses described that their FH was met with varying reactions from others. The reactions ranged from support and understanding to disbelief and criticism, and participants also described that others did not take their FH seriously. One participant with a food allergy diagnosis from conventional medicine gave an example of what she experienced as a lack of understanding. She said:

*Their opinion seemed to be (. . .) 'if you are not going to die from it, you can happily eat it'.*

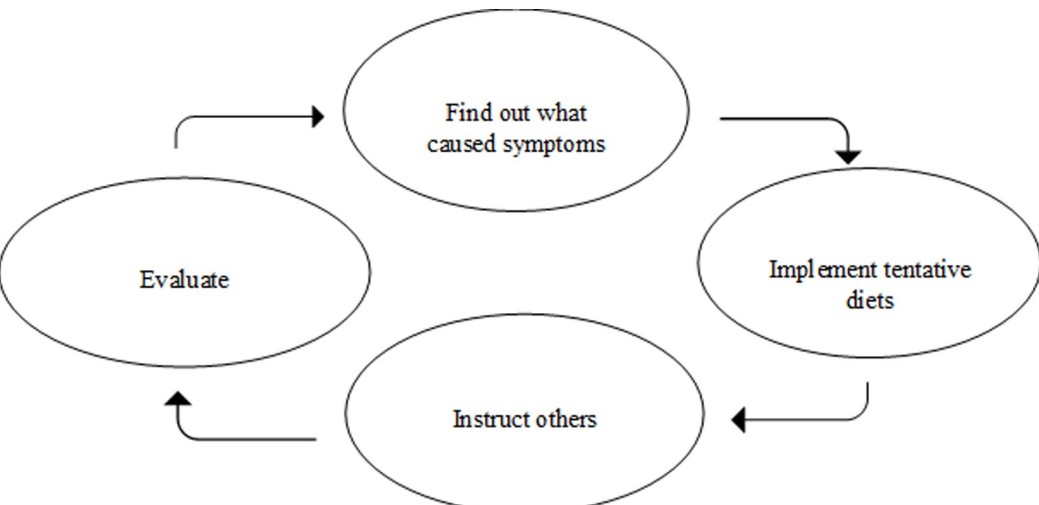

**Fig 1. A cycle of repeated tasks.** Women who had not clarified once and for all which food(s) caused symptoms, described carrying out repeated tasks to find out what caused symptoms, implement tentative diets, instruct others, and evaluate their self-management regimens.

Also, irrespective of diagnosis, participants described that negative reactions and stigma made them conceal their restricted diet and FH. One study participant with irritable bowel syndrome described such concealment:

*I do not talk about it or ask for adaptions. (...) Yes, it [another long-term condition] is a part of me that I cannot hide. (...) I can hide the FH a bit more.*

Some participants gave further explanations of why they concealed their restricted diet and FH. These reasons included not wanting to be the center of attention, the desire not to bother others, feeling that not being able to eat everything would be associated with weakness, and not wanting to be associated with alternative diets or with women who are obsessed with food. An interviewee without a FH diagnosis from conventional medicine said:

*When I learned that I had allergies back in the 80s (...) there was not the same focus on food that there is today (...). But now, like with gluten, I think even people without celiac disease avoid gluten because they have read that gluten is unhealthy (...) I do not want to be associated with those who have an almost unhealthy obsession with food.*

The participants who hid their FH described different strategies of concealment: Some participants said that they brought their own food and tried to find a way to eat it without being noticed. Others searched for something 'safe' to eat without revealing that they had restrictions, and some ingested foods that gave them symptoms to avoid revealing their FH.

## Discussion

In the present study, we explored the efforts people made to manage their FH. We used NPT, which is a conceptual framework formerly used to describe the self-management work of long-term conditions, as a tool to explore and describe these efforts. The women with FH who participated in the present study described a heightened vigilance when they ate food that others had prepared, and they carried out several tasks, some of which were time-consuming, to manage their FH. Participants also gave many descriptions of how they carried out these tasks themselves, with little help from others. Some also made efforts to conceal their FH, among other reasons because they experienced a lack of understanding from others, because they wanted to avoid bothering others with their dietary restrictions, or because they wanted to avoid being associated with alternative diets.

Participants who had clarified once and for all which food(s) caused symptoms, described that they could concentrate on carrying out a restricted diet, which could become routine. On the other hand, those who had not clarified once and for all what caused symptoms, described that they had to make repeated efforts to find out what caused symptoms, implement new tentative diets, instruct others in those tentative diets, and evaluate the management regimens. In other words, those who had not clarified what caused symptoms made efforts related to all four types of work included in the NPT. This indicates that diagnostic help from conventional medicine in clarifying what food(s) cause symptoms may not only lead to a reduction in symptoms and unnecessary dietary restrictions [18,35], but may also reduce the amount self-management work of FH.

Indeed, participants described carrying out several tasks, some of which were time-consuming, to manage their FH. Our interpretation of this is that managing a FH can imply a large workload and burden of treatment. Previous studies have reported that other long-term conditions may require large workloads [19,21,36]. Thus, FH shares aspects of other long-term conditions in the sense that it may entail a large self-management workload and burden of

treatment. Many people have concurrent long-term conditions, and this applies to a subgroup of women with FH as well [3,37–39]. According to May et al. [28] and Dobler et al. [25], the workload of managing one condition is part of the total burden of treatment, and the workload of managing one condition may reduce a person's capacity to do other tasks, including managing comorbidities. Therefore, May et al. [23] and Dobler et al. [25] argue that is it important for health care providers to be attentive to patients' total workload–the combination of things they need to do to manage all their conditions–and strive to keep this workload feasible by helping people reduce the amount of self-management work.

Previous studies have indicated that the NPT is a useful tool for understanding and describing the self-management work of long-term conditions [19,20]. In the present study, we also found the NPT to be a useful tool for exploring the self-management work of FH, including the self-management work of women without FH diagnoses from conventional medicine. However, while previous studies that used the NPT to explore self-management work reported that health care services assisted with tasks and appointments [19,20], participants in the present study described little contact with health care services and gave few examples of assistance from others. Our interpretation of this is that the participants did most of the self-management work of FH themselves. Some possible explanations for this may be that health care services in Norway have lacked the capacity and competence to diagnose FH [18], which may have led to an increased effort among the women to try to identify which food(s) cause symptoms by themselves. In addition, food and meal preparation are, to a large degree, the responsibility of women [40], which may have contributed to participants' reports that they did most of the grocery shopping and food preparation themselves.

Another possible explanation as to why participants seemed to do most of the self-management work of FH themselves may be related to the fact that some concealed their restricted diet and FH. Such concealment of a restricted diet has also been described in other studies [41,42]. Our participants' explanations of what made them conceal their restricted diet indicate that social aspects of food and the meal contribute to concealment. For example, some participants described that they concealed their restricted diet to avoid being associated with alternative diets. A possible interpretation of this is that mistaking peoples' FH conditions for 'alternative', voluntary diets can reduce the legitimacy of their restricted diet, as was suggested in a former study on celiac disease [43]. In addition, the number of people who have implemented an 'alternative' or 'healthy' diet has clearly increased in recent years [44]. On one hand, this increase can be seen as a corollary of the fact that it has become an obligation to take care of one's health, including through the adoption of a healthy diet [45–47]. However, on the other hand, many people are concerned that the common meal, taste, and tradition will be 'forgotten' and that exaggerated focus on nutrition and health can have negative consequences on health [46,47]. Thus, food and the meal can be considered a zone of disagreements and concerns, and our participants' wish to avoid being associated with alternative diets may be related to a wish not to be involved in this zone.

Participants also reported their wish not to bother others as a reason for carrying out the self-management work themselves and for concealing their FH. This is in line with studies that indicated that people do not want to burden others with their long-term conditions, since reliance on others may place strain on relationships and disrupt normal rules of reciprocity and mutual support [48–50]. Also Fischler [51] described reciprocity as important, and he argued that reciprocity is particularly important with regard to food and meals. For example, when a host offers a meal, the guest is supposed to appreciate the meal [52], and 'precautionary examinations' of food are perceived as unacceptable, childish, or as an expression of distrust towards the host [51]. Fischler also emphasized that food and the meal create and maintain social intimacy and social bonds, and that there are many social rules related to food and meals [51,53].

Thus, concealment of a restricted diet may consciously or intuitively be an attempt not to break the social rules of the meal and to attend to social relationships. However, regardless of why participants concealed their restricted diet and FH, they described making a concerted effort to do so. This effort can be understood as one part of the work participants carried out to manage their FH and as an appreciable part of their burden of treatment.

Participants also explicitly described that a lack of understanding and negative reactions from others made them conceal their restricted diet and FH, and participants described a heightened vigilance when eating foods other had prepared in order to avoid adverse reactions. Experiences with lack of understanding and heightened vigilance have also been described in previous studies on people with FH [54–56], and studies have indicated that lack of understanding and heightened vigilance may contribute to psychological distress and/or may have a negative impact on quality of life [54,55]. Moreover, Cummings et al. [54] emphasized that the daily management of FH conditions, which we in the present study understand as 'work', can have a psychosocial impact. This indicates that the management of FH may not only imply a considerable workload, but may also imply a psychosocial burden.

This study has some limitations. One is that the predefined themes may have led to less openness to other topics in the interviews [34]. The fact that only the first author read through the raw data can also be seen as a limitation, since two researchers may see more nuances in the data [57], and since coding of the raw data by two researchers can establish the "credibility" of the coding [58]. Further, the study only included women aged 39–67 years with relatively complex FH conditions, and inclusion of men, younger women, and persons with less complex FH conditions might have yielded somewhat different findings. Insight into the self-management work of these groups will require further research.

## Conclusions

The participants in the present study described carrying out several tasks, some of which were time-consuming, to manage their FH. This indicates that self-management of FH, like self-management of other long-term conditions, can imply a large workload and burden of treatment. While participants who had not clarified which food(s) caused symptoms made repeated efforts to find out what caused symptoms, implement new diets, and evaluate the management regimens, participants who had achieved such clarification could concentrate on carrying out a restricted diet. This indicates that diagnostic assistance from health care services could reduce the self-management work of FH, and this, in turn, may increase that capacity of individuals with FH to do other tasks, including managing comorbidities. Participants also described a heightened vigilance when they ate food others had prepared, and some made efforts to conceal their FH. These efforts should be considered part of the workload of managing FH and part of the burden of treatment.

## Acknowledgments

The authors would like to thank the interviewees and those who have helped us recruit interviewees.

## Author Contributions

**Conceptualization:** Monika Dybdahl Jakobsen, Aud Obstfelder, Tonje Braaten, Birgit Abelsen.

**Data curation:** Monika Dybdahl Jakobsen.

**Formal analysis:** Monika Dybdahl Jakobsen, Aud Obstfelder, Tonje Braaten, Birgit Abelsen.

**Methodology:** Monika Dybdahl Jakobsen, Aud Obstfelder, Tonje Braaten, Birgit Abelsen.

**Project administration:** Monika Dybdahl Jakobsen.

**Supervision:** Aud Obstfelder, Tonje Braaten, Birgit Abelsen.

**Validation:** Monika Dybdahl Jakobsen, Aud Obstfelder, Birgit Abelsen.

**Visualization:** Monika Dybdahl Jakobsen.

**Writing – original draft:** Monika Dybdahl Jakobsen.

**Writing – review & editing:** Monika Dybdahl Jakobsen, Aud Obstfelder, Tonje Braaten, Birgit Abelsen.

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
