## [Decision Letter · Decision Letter 0]

3 Dec 2020

PONE-D-20-28659

The self-management work of food hypersensitivity

PLOS ONE

Dear Dr. Jacobsen

Thank you for submitting your manuscript to PLOS ONE. After careful consideration, we feel that it has merit but does not fully meet PLOS ONE’s publication criteria as it currently stands. Therefore, we invite you to submit a revised version of the manuscript that addresses the points raised during the review process.

We look forward to receiving your revised manuscript.

Kind regards,

Patrizia Restani, Ph.D.

Academic Editor

PLOS ONE

Journal Requirements:

2. Please provide additional details regarding participant consent. In the ethics statement in the Methods and online submission information, please ensure that you have specified whether consent was informed.

4. Please list the name and version of any software package used for data analysis, alongside any relevant references, if applicable.

5.We note that you have indicated that data from this study are available upon request. PLOS only allows data to be available upon request if there are legal or ethical restrictions on sharing data publicly. For information on unacceptable data access restrictions, please see http://journals.plos.org/plosone/s/data-availability#loc-unacceptable-data-access-restrictions.

Additional Editor Comments (if provided):

Based on the reviewers' comments, the paper is interesting but requires some improvement. It is therefore necessary to review the paper taking into consideration the useful reviewers' suggestions

Reviewers' comments:

Reviewer's Responses to Questions

**Comments to the Author**

1. Is the manuscript technically sound, and do the data support the conclusions?

Reviewer #1: Partly

Reviewer #2: Yes

2. Has the statistical analysis been performed appropriately and rigorously? 

Reviewer #1: No

Reviewer #2: N/A

3. Have the authors made all data underlying the findings in their manuscript fully available?

Reviewer #1: Yes

Reviewer #2: No

4. Is the manuscript presented in an intelligible fashion and written in standard English?

Reviewer #1: Yes

Reviewer #2: Yes

5. Review Comments to the Author

Reviewer #1: Food hypersensitivity (FH) has received considerable attention in the scientific community in recent years, so a topic as this could be important for scientific community

Anyway, the quality of the article needs to be improved in order to be considered for publication in this journal

Some recommendations are indicated bellow

- it is not usual to find in introduction in one phrase a number of 12 references... with very low description

"Studies on FH have focused on several topics, including the epidemiological and medical aspects of FH [3-5], experiences of living with FH [6], and quality of life [7-15]"...

- the level of discussion needs to be also improved in order to add more references and to compare more deeply the obtained data

- innovation/new added data needs to be underlined in order to increase the importance of the performed study, as well as the associated drawbacks

- the conclusion needs to be rewritten underlining the most important findings and also the possible way to implement the obtained results

Reviewer #2: I really appreciated this paper describing the efforts made by people with food hypersensitivity, since the subjective perception of diseases is an aspect generally little considered respect to other more objective clinical aspects. Only few points need to be clarified by authors:

Line 99: authors should explain why only women were included in the study, despite this aspect is reported as one of the study limitations.

Line 149: Please give more details on how the pilot interview was carried out and developed.

Line 166: Please clarify whom “she” is referred to and why only one author was involved in the interpretation of data from the interviews. This is a critical aspect since several biases can occur.

Line 178: Please add standard deviation of participants' mean age.

Line 180: Did the authors find any differences between women with a recent FH diagnosis /symptoms development and those having had a long story of FH in managing the disease? Please add some sentences about this point.

In the “results” section a table or a graph reporting the main tasks and findings would make the results more clear.

6. PLOS authors have the option to publish the peer review history of their article (what does this mean?). If published, this will include your full peer review and any attached files.

Reviewer #1: No

Reviewer #2: No

---

## [Author Response · Author response to Decision Letter 0]

21 Jan 2021

Patrizia Restani, Ph.D.

Academic Editor

PLOS ONE

4 January 2021

Dear Editor

Thank you for very valuable comments and suggestions, which have contributed to important improvements of our article. We hope you find our changes of the text of sufficient quality and in line with your suggestions. Our answers to each of the reviewers’ comments are listed below.

Yours sincerely,

Monika Dybdahl Jakobsen

Our answers to Reviewer 1

Some recommendations are indicated bellow

- it is not usual to find in introduction in one phrase a number of 12 references... with very low description

"Studies on FH have focused on several topics, including the epidemiological and medical aspects of FH [3-5], experiences of living with FH [6], and quality of life [7-15]"...

Thank you for your valuable recommendations. We have now reformulated the sentence, and we have described the topics of previous research in more detail. The amendments can be found in lines 60-65.

- the level of discussion needs to be also improved in order to add more references and to compare more deeply the obtained data.

We have now compared our findings concerning the fact that some concealed their restricted diet and FH with previous studies, and we have added references. 

We have also added a comparison between our findings concerning the participants’ wish not to bother others and former studies, and we have added three references.

In addition, we have added information about the importance of being attentive to the workload of managing long-term conditions. References are also added, and the paragraph is of linguistic cases somewhat modified.

These amendments can be found in lines 355-358, 375-376 and 391-394.

- innovation/new added data needs to be underlined in order to increase the importance of the performed study, as well as the associated drawbacks.

With all revisions made, we hope that our revised article more clearly describe what the study contributes with in the introduction and discussion. We have also amended the discussion on the study’s limitations, and these amendments can be found in lines 416-419. We would very much like to keep the paper as short as possible, and hope that the study contributions and limitations are sufficiently described.

- the conclusion needs to be rewritten underlining the most important findings and also the possible way to implement the obtained results. 

We have now rewritten the conclusion and have included the most important findings. We have also written in a clearer way why our findings indicate that diagnostic help from health care services may be significant.

The amendments can be found in lines 425-436.

Our answers to Reviewer 2 

I really appreciated this paper describing the efforts made by people with food hypersensitivity, since the subjective perception of diseases is an aspect generally little considered respect to other more objective clinical aspects. Only few points need to be clarified by authors:

Line 99: authors should explain why only women were included in the study, despite this aspect is reported as one of the study limitations.

Thank you for your important comments. We agree that subjective perception has been generally little considered. The inclusion of women only was a result of the fact that the present study is part of a project that only includes women. We have included this information in the text, and it can be found in lines 102-104.

Line 149: Please give more details on how the pilot interview was carried out and developed.

We have now included more details on how the pilot interview was carried out and developed. This information can be found in lines 153-156.

Line 166: Please clarify whom “she” is referred to and why only one author was involved in the interpretation of data from the interviews. This is a critical aspect since several biases can occur.

Thank you for this very important comment. We have rephrased the sentence to clarify that “she” refers to the first author. Your comment also pointed to the fact that we have not been precise enough in the description of the coauthors’ role in the process of analyzing the data. The coauthors participated in the process of analyzing the data, however only the first author read the raw data. This is now specified. 

As a result of your comment on biases, we have added a clause (with reference) to the text concerning limitations. In this clause we write that two researchers’ coding of the raw data can contribute to establish the credibility of the coding.

These amendments can be found in lines 171-177 and lines 416-419. 

Line 178: Please add standard deviation of participants' mean age.

We agree that a presentation of the variation in age will contribute to a better description of the sample. However, in this qualitative study with a small sample, we suggest to present how many women belonged to each age group (instead of standard deviation), since we mean that this may contribute to a clear/thorough description of the variation in age. 

You will find these amendments in lines 186-188. 

Line 180: Did the authors find any differences between women with a recent FH diagnosis /symptoms development and those having had a long story of FH in managing the disease? Please add some sentences about this point.

We had not written clear enough that all women described having had FH symptoms for a long time (years or decades). We have now accentuated this, and the amendment can be found in lines 188-189.Moreover, we did not find any differences between women who have had FH symptoms for years and those who had an even longer story of FH. The clear difference was between 1) those who had clarified once and for all which food(s) caused symptoms and 2) women who had not achieved such clarification.

In the “results” section a table or a graph reporting the main tasks and findings would make the results more clear. 

One central finding was that those who had not clarified once and for all which food(s) caused symptoms, described a cycle of repeated tasks to find out what caused symptoms, implement new tentative diets, instruct others in tentative diets, and evaluate these tentative self-management regimens. We have now included a figure (figure 1) that illustrates this finding. The figure title and legend can be found in lines 290-292, and the figure is uploaded separately, which is in line with PLOS ONE’s style requirements.

---

## [Decision Letter · Decision Letter 1]

22 Feb 2021

The self-management work of food hypersensitivity

PONE-D-20-28659R1

Dear Dr. Jakobsen

We’re pleased to inform you that your manuscript has been judged scientifically suitable for publication and will be formally accepted for publication once it meets all outstanding technical requirements.

Kind regards,

Patrizia Restani, Ph.D.

Academic Editor

PLOS ONE

Additional Editor Comments (optional):

All comments have been considered and changes are satisfactory

Reviewers' comments:

Reviewer's Responses to Questions

**Comments to the Author**

1. If the authors have adequately addressed your comments raised in a previous round of review and you feel that this manuscript is now acceptable for publication, you may indicate that here to bypass the “Comments to the Author” section, enter your conflict of interest statement in the “Confidential to Editor” section, and submit your "Accept" recommendation.

Reviewer #1: All comments have been addressed

Reviewer #2: All comments have been addressed

2. Is the manuscript technically sound, and do the data support the conclusions?

Reviewer #1: Yes

Reviewer #2: Yes

3. Has the statistical analysis been performed appropriately and rigorously? 

Reviewer #1: Yes

Reviewer #2: N/A

4. Have the authors made all data underlying the findings in their manuscript fully available?

Reviewer #1: Yes

Reviewer #2: Yes

5. Is the manuscript presented in an intelligible fashion and written in standard English?

Reviewer #1: Yes

Reviewer #2: Yes

6. Review Comments to the Author

Reviewer #1: Food hypersensitivity (FH) has received considerable attention in the scientific community in recent years, so this article could be important for PlosOne readers

The supplementary comments and answers have contributed to improvements of the article

Reviewer #2: (No Response)

7. PLOS authors have the option to publish the peer review history of their article (what does this mean?). If published, this will include your full peer review and any attached files.

Reviewer #1: No

Reviewer #2: No

---

## [Editor Report · Acceptance letter]

25 Feb 2021

PONE-D-20-28659R1 

The self-management work of food hypersensitivity 

Dear Dr. Jakobsen:

I'm pleased to inform you that your manuscript has been deemed suitable for publication in PLOS ONE. Congratulations! Your manuscript is now with our production department. 

Kind regards, 

on behalf of

Professor Patrizia Restani 

Academic Editor

PLOS ONE